# The Relationship Between the Distribution of Neural Network Weights and Model Accuracy Using Benford's Law

## Abstract

**Context:** Benford's Law describes the distribution of atypical patterns of numbers. It focuses on the occurrence of the first digit in a natural population of numbers. When these numbers are divided into nine categories based on their first digit, the largest category consists of numbers that start with 1, followed by those starting with 2, and so on. **Objective:** Each neuron in a Neural Network (NN) holds a mathematical value, often referred to as a weight, which is updated according to certain parameters. This study explores the Degree of Benford's Law Existence (DBLE) within Convolutional Neural Networks (CNNs). Additionally, the experiment investigates the correlation between the DBLE and NN's accuracy. **Methods:** A (CNN) is subjected to testing 15 times using various datasets and hyperparameters. The DBLE is calculated for each CNN variation, and the correlation between the CNN's performance and DBLE is examined. To further explore the presence of Benford's Law in CNN models, nine transfer learning models are also tested for. **Results:** The experiment suggests: 1) Benford's Law is observed in the weights of neural networks, and in most cases, the DBLE increases as the training progresses. 2) It is observed that models with significant differences in performance tend to demonstrate relatively high divergence in DBLE.

## 1 Introduction

Benford's Law is a phenomenon that occurs in natural distributions of numbers worldwide. This law states that, typically, within a population of numbers, there are more numbers that begin with smaller digits than those that begin with larger digits. For example, in a given natural population of numbers, there will generally be more numbers that start with 1 compared to those that start with 2, and so on. Figure 1(a) visually represents this phenomenon. The first bar (labeled as 1) represents the proportion of numbers with a first digit of 1, the second bar (labeled as 2) represents the proportion of numbers with a first digit of 2, and so on. It can be observed that the group with numbers starting with 9 is the smallest, while the group starting with 1 is the largest.

Benford's Law was initially discovered accidentally in 1881 by astronomer Simon Newcomb. He observed that the early pages of a logarithm book were more worn compared to the later pages. This logarithm book contained the logarithms of numbers used by scientists for multiplication in the 19th century, prior to the advent of fast and inexpensive calculators. The book was organized in a manner where numbers starting with the digit 1 were listed at the beginning, followed by those starting with the digit 2, and so on. In 1939, *Frank Benford*, a physicist at General Electric, also stumbled upon the same law discovered by Newcomb. He further explored this law by conducting tests on various large datasets of numbers. Although Newcomb was the first to notice this phenomenon, the law was named in honor of Frank Benford due to his extensive investigation and contributions.

Benford's law may be also represented as a mathematical formula that indicates the most accurate value, percentage, or frequency for each category of numbers. This percentage or the frequency is calculated by Equation 1. For instance, the percentage of a set of numbers that starts with digit 3 would be $12.49\% = \log_{10}(3+1) - \log_{10}(3)$

Benford's Law can also be expressed as a mathematical formula that determines the expected value, percentage, or frequency for each category of numbers. This percentage or frequency is calculated using Equation 1. For example, the percentage of a set of numbers starting with the digit 3 would be 12.49%, which is obtained by evaluating the expression $\log_{10}(3+1) - \log_{10}(3)$.

$$P(d) = \log_{10}(d+1) - \log_{10}(d) \tag{1}$$

Figure 1(a) depicts the expected percentage distribution according to Benford's Law with the following categories: **[0.301, 0.176, 0.125, 0.097, 0.079, 0.067, 0.058, 0.051, 0.046]**.

In this experiment, the designated list is referred to as **BenIdeal**. Benford's Law has been observed in various real-world and natural numerical populations. For instance, the populations of different countries are expected to follow Benford's Law, although the actual percentages of each category may not precisely align with Equation 1 and Figure 1(a).

Figure 1(b) [1] illustrates the percentage distribution of each category based on the populations of different countries. For example, approximately 29% of all countries' populations start with the digit 1, including countries like India (1,417,173,173) and Cyprus (1,251,488).

To determine whether a given population of numbers adheres to Benford's Law, a list of 9 values is created based on their first digit. This list is then compared against the list generated by Equation 1. The Degree of Benford Law Existence (DBLE) is calculated using linear regression as a means of comparison. The subsequent section delves into further details.

## 1.1 Machine Learning

Machine Learning, as a sub-field of Artificial Intelligence, is involved in the development and creation of techniques such as statistical and neural network algorithms that enable computer systems to learn from existing data in order to predict or make decisions.

Neural Network (NN) algorithms are a subset of Machine Learning algorithms that are usually comprised of a set of layers holding a number of neurons, and these neurons are connected to each other in some fashion.

The quality of any NN algorithm may depend on several factors, such as: 1) the amount of data used for training, 2) the model's hyperparameters, 3) the model's architecture, and so on. One common approach to measure the quality of Neural Networks is to count the number of correct predictions and divide it by the total number of predictions: Accuracy = (Number of Correct Predictions) / (Total Number of Predictions).

In this work, a new hypothesis is proposed and examined as a complementary method to compare the quality of two or more models without running predictions. This hypothesis aims to explore if there is any relationship between a model's accuracy and the distribution of all the model's weights.

## 2 Related Works

Benford's Law has applications in various fields, including cyber-security and finance. For instance, in their work (Jianu & Jianu, 2021), Jianu analyzed the reliability of financial information from the perspective of Benford's Law. In a separate study, Dutta et al., (Dutta et al., 2023) delved into the causes of the collapse of "Silicon Valley Bank" by applying Benford's Law to ascertain if there was any alteration or evidence of data manipulation. Their findings revealed suspicious discrepancies between the bank's financial data and the distribution of digits, indicating the presence of fraud. Similarly, in another study by Vicic et al., (Vičič & Tošić, 2022), Benford's Law was employed to detect anomalies, potentially indicating fraudulent behavior, in cryptocurrency transactions.

The application of Benford's Law extends beyond the realms of finance and cyber security to encompass various other fields. Schumm et al., (Schumm et al., 2023), for instance, employed Benford's Law to identify unusual anomalies in retractions. They propose several valuable methods that can aid in identifying questionable research practices, particularly instances of data or results fabrication, across domains such as social science, medicine, and other scientific research. In two distinct studies, Varga (Varga, 2021a) and (Varga, 2021b) employed Benford's Law for no-reference quality assessment (NR-IQA, NR-VQA) of images and videos. Their research showcased that first-digit distributions extracted from various transformed image and video domains can function as quality-aware features, establishing a meaningful correlation with perceptual quality scores.

---

[1] https://worldpopulationreview.com/countries

Furthermore, Benford's Law finds application in the natural realm. Proger et al., (Pröger et al., 2021), for instance, conducted a study involving 1132 weekly logbooks from 20 wild red deer individuals over a span of more than four years. The study focused on recording the lengths of the deer's walks. The collected data exhibited a distribution of first digits resembling Benford's Law. However, the adherence to Benford's Law was not perfect due to limited evidence, as explained in their study. This departure from the law might be attributed to human intervention, as the deer tended to avoid human presence. Notably, the study area was a tourist destination in both winter and summer. In another study by Kolias et al., (Kolias, 2022), Benford's Law was applied to daily COVID-19 cases across all countries. The study reported that Bulgaria, Croatia, Lithuania, and Romania aligned well with Benford's distribution, while Denmark, Ireland, and Greece exhibited the greatest deviations from Benford's distribution.

Benford's Law, when combined with Machine Learning, has applications across various fields. In their work, Elena Badal-Valero and colleagues (Badal-Valero et al., 2018) assessed the application of logistic regression, decision trees, neural networks, and random forests, along with Benford's Law, to discover patterns of money laundering in a Spanish court case involving criminals. Fletcher and Boritz (Lu & Boritz, 2005) employed an unsupervised learning method in combination with Benford's Law to detect fraud and abuse in health insurance claims using real health insurance data. Their technique demonstrates higher precision compared to the traditional Benford approach.

The successful application of Benford's Law in various fields, as discussed above, serves as motivation to take a step forward and apply Benford's Law in the context of Machine Learning, specifically in relation to neural network weights, for the very first time.

## 3   Methodology

One of the main objectives of this study is to investigate the Degree of Benford's Law Existence (DBLE) within Convolutional Neural Networks (CNNs) and explore its potential correlation with network performance, such as accuracy. A CNN is composed of multiple layers, each containing a set of neurons arranged in arrays or multidimensional arrays. Neurons store numerical values represented as decimal numbers.

The performance of neural networks depends on various factors, including hyperparameters, optimizers (Feurer & Hutter, 2019) (Yu & Zhu, 2020), network architecture (Benardos & Vosniakos, 2007), learning rate, the size of the training dataset (Foody et al., 1995), and the quality of the training data (Kavzoglu, 2009).

### 3.1   The Degree of Benford Law Existence

In order to explore the correlation between neural network (CNN) performance and the Degree of Benford Law Existence (DBLE), the weights of each layer in the neural network are collected as a collection of decimal numbers, and the DBLE is computed.

Benford's Law is defined by Equation 1, where $d$ represents a non-zero and non-negative digit. $P(d)$ signifies the expected ideal percentage of numbers starting with the digit $d$. In a neural network, the weights of neurons are real numbers, such as $0.001208$ or $-0.023$. However, the original definition of Benford's Law is based on natural numbers (positive integers starting from 1). To align it with the weights of neurons, each real number is converted to a natural number by eliminating all zeros between the decimal point and the first non-zero digit. The resulting number is then multiplied by 10 and the absolute value of the integer part is taken. For instance, $0.001208$ becomes 1, and $-0.023$ becomes 2. As a result, for any given set of real numbers, it becomes possible to compute the percentage of numbers starting with 1, 2, and so on up to 9. Ultimately, this yields a list of 9 percentages, where the sum totals 1.0. In this study, this list is referred to as ***BenList***. To determine the extent to which Benford's Law exists (DBLE) in a given population of real numbers, the ***BenList*** is calculated and compared to the ***BenIdeal***. Various approaches can be employed to assess the similarity between two distributions of numbers, (e.g., ***BenList*** and ***BenIdeal***). One of the approaches can be the use of the *chi-square test*; however, Kossovsky et al., (Kossovsky, 2021) discussed the inappropriateness of the *chi-square test* in the context of Benford's Law in their work.

In an ideal scenario, the ***BenList*** would be identical to the ***BenIdeal***. Consequently, plotting the ***BenList*** against the ***BenIdeal*** would result in a straight line, as depicted in Figure 1(c). Conversely, a different ***BenList*** would yield a distinct line when plotted against the ***BenIdeal***. For instance, in the case of country populations, the resulting line would resemble Figure 1(d).

| Absolute Value of PCC | Interpretation |
|---|---|
| 0.00–0.10 | Negligible correlation |
| 0.10–0.39 | Weak correlation |
| 0.40–0.69 | Moderate correlation |
| 0.70–0.89 | Strong correlation |
| 0.90–1.00 | Very strong correlation |

Table 1: Interpretation of PCC

Hence, one can infer that the straighter the line, the greater the similarity between the **BenList** and **BenIdeal**. The line's straightness is measured using linear regression techniques. The Pearson Correlation Coefficient (PCC) (Sedgwick, 2012) is a real value ($r$) that falls within the range of $[-1, 1]$. A value of $-1$ indicates maximum negative linear correlation, while a value of 1 indicates maximum positive linear correlation between two variables. Values close to 0 indicate a weaker linear correlation, and a value of 0 signifies no correlation. $r$ can be calculated with the below formula.

$$r = \frac{\sum (X - \bar{X})(Y - \bar{Y})}{\sqrt{\sum (X - \bar{X})^2 \times (Y - \bar{Y})^2}}$$

Schober et al.

In this study, the implementation of *Pearsonr* from the *Scipy* module[2] is utilized. In addition to the Pearson Correlation Coefficient ($r$), the *Pearsonr* function also calculates the *p*-value. The *p*-value is employed to assess the rejection of the null hypothesis ($H_0$), which states that there is no relationship between two distributions of values. A *p*-value less than 0.05 is commonly accepted as evidence to reject the null hypothesis, indicating a linear relationship between the two distributions. In the context of this work, the **BenIdeal** and **BenList** are treated as two distributions of values, each comprising 9 values. The $r$(PCC) and *p*-value are then calculated to quantify the Degree of Benford Law Existence (DBLE) across the two datasets. For a given deep learning model, the **BenList** is computed for the weights of each layer individually. Subsequently, the $r$(PCC) and *p*-value are computed between the **BenList** of the layer and the **BenIdeal**, as illustrated in Figure 2. It is important to note that the $r$(PCC) and *p*-value are calculated for each layer. Therefore, the final results of the $r$(PCC) and *p*-value for each model are averaged across all layers.

---

[2]https://scipy.org/

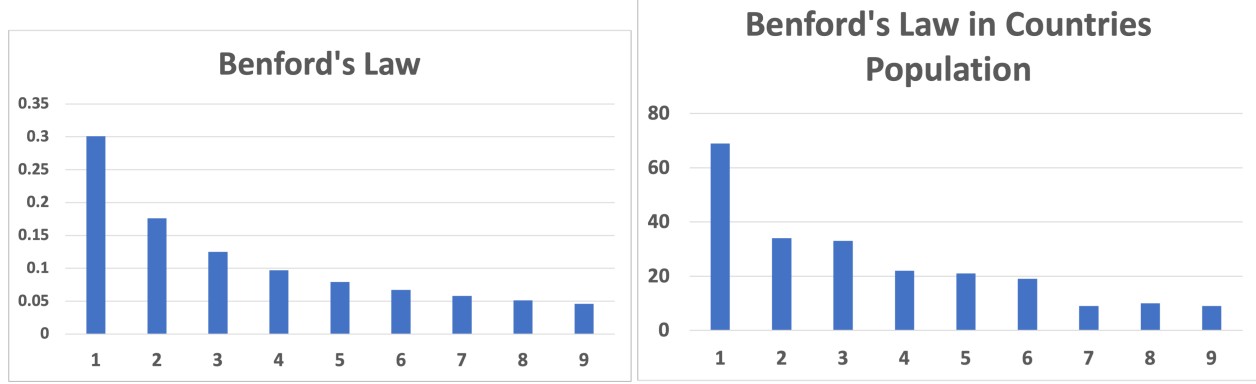

(a) The precise percentage of each category of numbers according to Equation 1.

(b) The percentage of each category of numbers from the full list of countries population in 2023

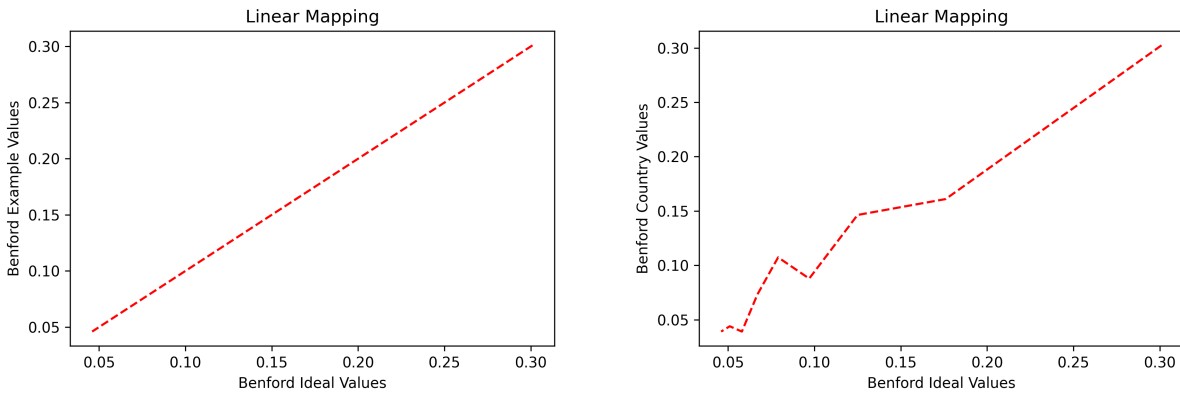

(c) The mapping of two identical Benford Distributions that constitute a straight line.

(d) The mapping of *BenList* and *BenIdeal*.

Figure 1: Plotting the *BenList* against *BenIdeal*.

This experiment aims to investigate the following questions:

1. Do the weights of neurons in Convolutional Neural Networks conform to Benford's Law?

2. How do the weights of neurons change, from the perspective of Benford's Law, as the model undergoes training?

## 4  Experiments

To investigate the aforementioned questions, several experiments are performed as follows:

1. Binary Image Classification with 1000 images using 5 different optimizers, while keeping all other features the same.

2. Binary Image Classification with the same 1000 images as above using Adam optimizer and 5 different learning rates, while keeping all other features the same.

3. Binary Image Classification with different 600 images using 5 different optimizers, while keeping all other features the same.

4. Additionally, the DBLE on the weights of 9 Transfer Learning models is analyzed without any extra training.

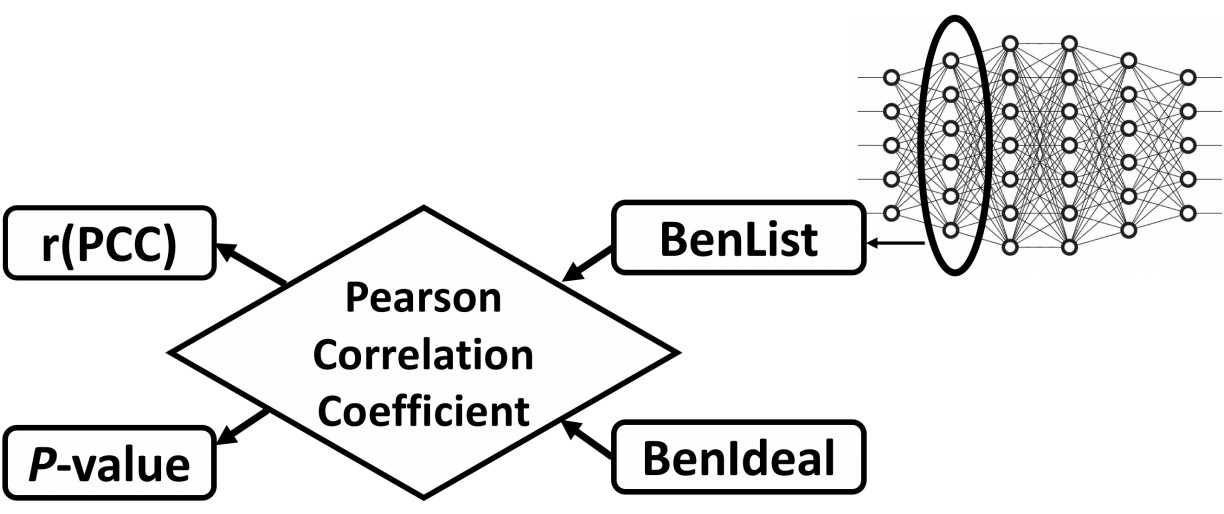

Figure 2: The flowchart of the process of measuring the Degree of Benford Law Existence (DBLE) in a neural network layer.

| Layer (type) | Output Shape | Param # |
|---|---|---|
| conv2d_90 (Conv2D) | (None, 198, 198, 32) | 896 |
| activation_120 (Activation) | (None, 198, 198, 32) | 0 |
| max_pooling2d_90 (MaxPooling2D) | (None, 99, 99, 32) | 0 |
| conv2d_91 (Conv2D) | (None, 97, 97, 32) | 9248 |
| activation_121 (Activation) | (None, 97, 97, 32) | 0 |
| max_pooling2d_91 (MaxPooling2D) | (None, 48, 48, 32) | 0 |
| conv2d_92 (Conv2D) | (None, 46, 46, 64) | 18496 |
| activation_122 (Activation) | (None, 46, 46, 64) | 0 |
| max_pooling2d_92 (MaxPooling2D) | (None, 23, 23, 64) | 0 |
| flatten_30 (Flatten) | (None, 33856) | 0 |
| dense_60 (Dense) | (None, 64) | 2166848 |
| activation_123 (Activation) | (None, 64) | 0 |
| dropout_30 (Dropout) | (None, 64) | 0 |
| dense_61 (Dense) | (None, 1) | 65 |

Table 2: The summary and the layers of the applied Neural Network. Layers are ordered from the top of the table to the bottom.

The first three experiments involve binary image classification learning on a deep neural network model, with the details shown in Table 2. The last experiment, on the other hand, does not involve any learning. Instead, a set of Transfer Learning models is experimented without any further training.

In all of these experiments, the Degree of Benford Law's Existence (DBLE) among the neurons' weights is measured using two metrics: 1) Linear Regression (The Pearson Correlation Coefficient, PCC) between each ***BenList*** and ***BenIdeal***. 2) $p$-value. The first metric, r(PCC), measures the relevance between the two variables (i.e., ***BenList*** and ***BenIdeal***), while the second metric tests whether the null hypothesis is rejected between ***BenList*** and ***BenIdeal***.

## 4.1 Model Optimizer

An optimizer in a Neural Network is an algorithm that adjusts the neurons' weights. The purpose of an optimizer is to improve the efficiency of the model, which translates to increasing the model's accuracy and reducing its loss. Several different optimizers have been introduced in the literature, including Gradient Descent, Stochastic Gradient Descent, Adagrad, RMSProp, Adam, etc. Choosing an appropriate optimizer is an important step in the learning process, taking

into account factors such as the task at hand (e.g., classification or regression) and the type of problem (e.g., binary or multi-class classification) (Smith, 2018).

Typically, when setting up a Neural Network, weights are initially established through random generation or by adhering to predefined distribution mechanism. These weights are then further adjusted epoch to epoch based on the specific mechanism articulated within the optimizer. This process of weight adjustment, among other factors, heavily depends on the architecture and functionality embedded within the optimizer. Therefore, the investigation into how the DBLE changes with distinct optimizers aims to investigate into potential correlations between optimizer and variations in DBLE values. This exploration can shed light on how different optimization techniques influence the existence of Benford's law withing the distribution of weights in a Neural network.

In this experiment, the relationship between optimizers and the DBLE among the neurons' weights is investigated using r(PCC) and $p$-value, as discussed earlier. The experiment is conducted twice on two open-source datasets (Bansal) and (cov), as follows:

### 4.1.1 First DataSet

The first dataset (Bansal) contains images classified into six categories: *buildings*, *forest*, *glacier*, *mountain*, *sea*, and *street*. Since this experiment focuses on binary classification, a random pair of forest and glacier images was selected for the actual experiment.

The details of the model's architecture are shown in Table 2. Each class of images contains 500 images, resulting in a total of 1000 images. Cross-validation with a 20% test dataset and 5 folds is applied to the learning process, with 800 images for training and 200 images for testing. Five optimizers are selected for this experiment: *Adam*, *Adadelta*, *Adagrad*, *Adamax*, and *RMSprop*. The following metrics are measured:

1. Test Accuracy (Average and Maximum).

2. Training Accuracy (Average and Maximum).

3. Test Loss (Average and Minimum).

4. Training Loss (Average and Minimum).

5. *r* (PCC) (Average and Maximum).

6. *P*-Value (Average and Minimum).

To avoid bias from the random division of test/train datasets, the experiment is performed using a five-fold cross-validation technique, where the test dataset size is set at 20% of the entire dataset.

Figure 3 (left table) displays the results of the five optimizers. The results in each row are color-coded from dark green to dark red, with dark green indicating the best result and dark red indicating the worst result. This allows easy comparison of the optimizers' performance in each metric. To examine whether higher DBLE values (r(PCC) and p-value) tend to be associated with better performance, one can observe each column to see the similarity between the colors associated with r(PCC) and p-value and the colors of other metrics (Accuracy and Loss). As shown in Figure 3, Adadelta appears to have the worst values for all factors, including DBLE (r(PCC) and p-value), while Adagrad has, on average, the second-worst values. On the other hand, Adam exhibits, on average, the best values, while Adamax and RMSprop fall somewhere in between. For better comparison, all values are ranked within each row and then color-coded in the right table of Figure 3.

To provide a more detailed view, these features (Accuracy, Loss, r(PCC), p-value) are graphed over 300 epochs and displayed in Figure 4. Since Adadelta and Adagrad have similar r(PCC) and p-value values, they are visualized separately with a lower scale to observe their evolution over 300 epochs (see Figures 4(g) and 4(h)). It is important to note that all values are averaged over the five folds.

| Optimizers => | Adam | Adadelta | Adagrad | Adamax | RMSprop | | Adam | Adadelta | Adagrad | Adamax | RMSprop |
|---|---|---|---|---|---|---|---|---|---|---|---|
| Ave r (PCC) | 0.87830 | 0.77701 | 0.77754 | 0.84213 | 0.85485 | | 1 | 5 | 4 | 3 | 2 |
| Ave P-Value | 0.01296 | 0.03030 | 0.03009 | 0.01607 | 0.02169 | | 1 | 5 | 4 | 2 | 3 |
| Ave test Acc | 0.98022 | 0.90799 | 0.96467 | 0.98312 | 0.97108 | | 2 | 5 | 4 | 1 | 3 |
| Ave train Acc | 0.99181 | 0.88781 | 0.95153 | 0.98970 | 0.98248 | | 1 | 5 | 4 | 2 | 3 |
| Ave test Loss | 0.10050 | 0.33369 | 0.11901 | 0.05732 | 0.15468 | | 2 | 5 | 3 | 1 | 4 |
| Ave train Loss | 0.02324 | 0.35595 | 0.14545 | 0.03073 | 0.06621 | | 1 | 5 | 4 | 2 | 3 |
| Max r (PCC) | 0.88440 | 0.77735 | 0.77790 | 0.86043 | 0.88890 | | 2 | 5 | 4 | 3 | 1 |
| Min P-Value | 0.01095 | 0.03006 | 0.02981 | 0.01287 | 0.00897 | | 2 | 5 | 4 | 3 | 1 |
| Max test Acc | 0.99500 | 0.97800 | 0.98400 | 0.99500 | 0.99400 | | 1 | 5 | 4 | 1 | 3 |
| Min test Loss | 0.02190 | 0.12544 | 0.07296 | 0.02250 | 0.01922 | | 2 | 5 | 4 | 3 | 1 |
| Max train Acc | 1.00000 | 0.96150 | 0.97375 | 1.00000 | 0.99975 | | 1 | 5 | 4 | 1 | 3 |
| Min train Loss | 0.00066 | 0.15995 | 0.09251 | 0.00225 | 0.00080 | | 1 | 5 | 4 | 3 | 2 |

Figure 3: Left Table: A quantitative comparison analysis was conducted on five optimizers. Values in each row are color-coded, with dark green representing the best value and dark red representing the worst value. The right table presents a ranking comparison of the five optimizers. Values in each row are ranked based on their goodness and then color-coded, with dark green representing the best value and dark red representing the worst value.

### 4.1.2 Second DataSet

The second dataset (cov) consists of four categories of images: *Confirmed Covid chest X-Ray*, *Confirmed Pneumonia chest X-Ray*, *Confirmed Tuberculosis chest X-Ray*, and *Normal X-Ray*. For binary classification, the experiment focuses on the "Confirmed Pneumonia chest X-Ray" and "Normal X-Ray" categories.

The model's architecture details are presented in Table 2. Each image category contains 300 images, resulting in a total of 600 images. Cross-validation with a test dataset size of 20% and 5 folds is applied, with 480 images for training and 120 images for testing. The same five optimizers (*Adam*, *Adadelta*, *Adagrad*, *Adamax*, and *RMSprop*) are selected, and the metrics mentioned earlier are measured.

Figure 5 displays the results of the five optimizers. The values in each row are color-coded, following the same scheme as in the previous experiment. This allows us to observe the best and worst performing optimizers for each metric. To determine if higher DBLE tends to have better performance, we can compare the colors associated with r(PCC) and *p*-value with the colors of other metrics (Accuracy and Loss) in each column. As shown in Figure 5, *Adadelta* appears to have the worst values across all factors, including r(PCC) and *p*-value, while *Adagrad* has the second worst values on average. Conversely, *Adam* exhibits the best values on average, with *Adamax* and *RMSprop* falling in between. To facilitate comparison, all values are ranked within each row and color-coded in the right table of Figure 5.

For a more detailed view, the features (Accuracy, Loss, r(PCC), *p*-value) are graphed over 300 epochs and displayed in Figure 6. Since *Adadelta* and *Adagrad* have similar r(PCC) and *p*-value, they are visualized separately with a lower scale to observe their evolution over 300 epochs in Figure 6(h).

### 4.2 Model Learning-Rate

One of the most critical hyperparameters in a deep neural network is the *Learning Rate* (Goodfellow et al., 2016). The learning rate determines the rate at which the model learns and converges to the global minimum loss. It is a tuning parameter typically set between 0 and 1. A lower learning rate requires more time to train as it takes smaller steps towards the global minimum loss, while a higher learning rate results in faster changes and potentially requires fewer epochs. However, using learning rates that are too large can lead to sub-optimal weight values, while very small learning rates may take a long time to converge or even get stuck before reaching the global minimum loss (Li et al., 2019).

In this experiment, the relationship between the learning rate and the Degree of Benford Law's Existence (DBLE) among neurons' weights is investigated using the r(PCC) and *p*-value metrics as discussed in section 3.

The experiment is conducted on the same open-source dataset (Bansal), and a new pair of image categories, *mountain* and *sea* is randomly selected for the task. The experiment is repeated five times with different learning rates: 0.0001,

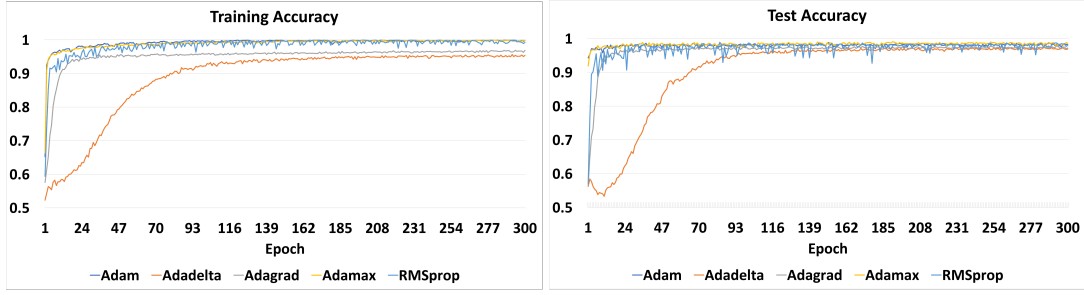

(a) The accuracy of Training dataset over 300 epochs for *forest* and *glacier* datasets. Note that all values are averaged over 5 folds.

(b) The accuracy of the Test dataset over 300 epochs for *forest* and *glacier* datasets. Note that all values are averaged over 5 folds.

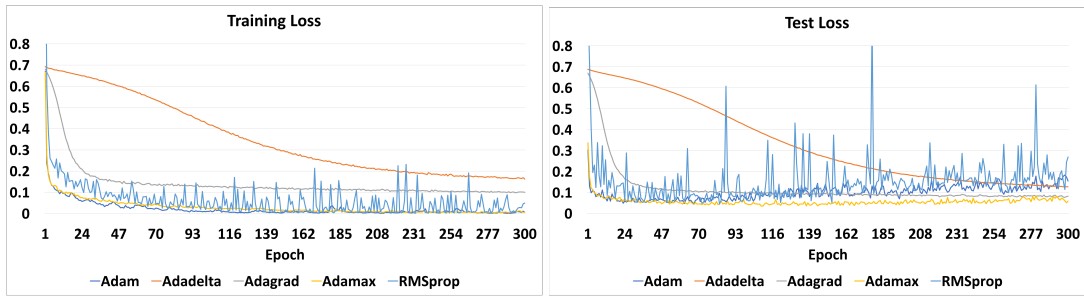

(c) The Loss of Training dataset over 300 epochs for *forest* and *glacier* datasets. Note that all values are averaged over 5 folds.

(d) The Loss of Test dataset over 300 epochs for *forest* and *glacier* datasets. Note that all values are averaged over 5 folds.

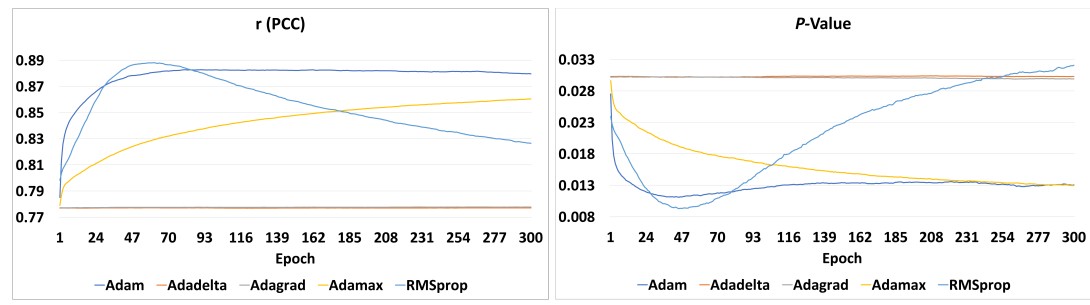

(e) The evolution of *r*-PCC on *BenList* and *BenIdeal* over 300 epochs for *forest* and *glacier* datasets. Note that all values are averaged over 5 folds.

(f) The evolution of *p*-Value on *BenList* and *BenIdeal* over 300 epochs for *forest* and *glacier* datasets. Note that all values are averaged over 5 folds.

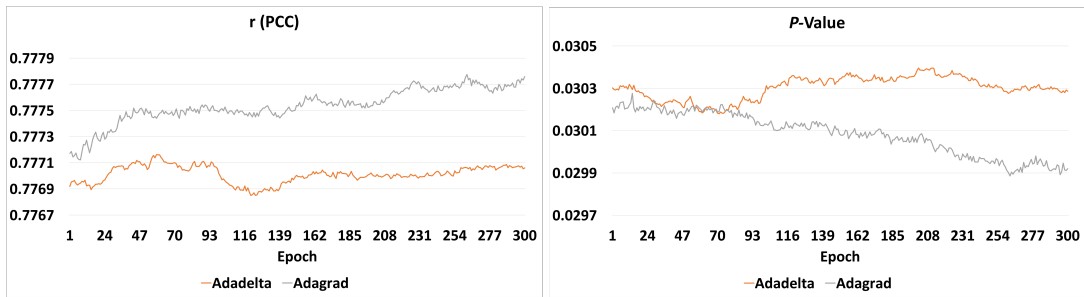

(g) Adagrad and Adadelta: The evolution of *r*-PCC on *BenList* and *BenIdeal* over 300 epochs for *forest* and *glacier* datasets. Note that all values are averaged over 5 folds.

(h) Adagrad and Adadelta: The evolution of *p*-Value on *BenList* and *BenIdeal* over 300 epochs for *forest* and *glacier* datasets. Note that all values are averaged over 5 folds.

Figure 4: The efficiency details of five optimizers for binary image classification on *forest* and *glacier* datasets.

| Optimizers => | Adam | Adadelta | Adagrad | Adamax | RMSprop | | Adam | Adadelta | Adagrad | Adamax | RMSprop |
|---|---|---|---|---|---|---|---|---|---|---|---|
| Ave r (PCC) | 0.8360 | 0.7777 | 0.7782 | 0.8129 | 0.8561 | | 2 | 5 | 4 | 3 | 1 |
| Ave P-Value | 0.0159 | 0.0296 | 0.0295 | 0.0204 | 0.0144 | | 2 | 5 | 4 | 3 | 1 |
| Ave test Acc | 0.9254 | 0.6462 | 0.7812 | 0.9208 | 0.9117 | | 1 | 5 | 4 | 2 | 3 |
| Ave train Acc | 0.9407 | 0.5941 | 0.7538 | 0.9268 | 0.9304 | | 1 | 5 | 4 | 3 | 2 |
| Ave test Loss | 0.1987 | 0.6807 | 0.5342 | 0.2143 | 0.2280 | | 1 | 5 | 4 | 2 | 3 |
| Ave train Loss | 0.1502 | 0.6814 | 0.5391 | 0.1968 | 0.1733 | | 1 | 5 | 4 | 3 | 2 |
| Max r (PCC) | 0.8490 | 0.7778 | 0.7784 | 0.8249 | 0.8728 | | 2 | 5 | 4 | 3 | 1 |
| Min P-Value | 0.0137 | 0.0295 | 0.0293 | 0.0175 | 0.0111 | | 2 | 5 | 4 | 3 | 1 |
| Max test Acc | 0.9783 | 0.8083 | 0.9133 | 0.9767 | 0.9800 | | 2 | 5 | 4 | 3 | 1 |
| Min test Loss | 0.0769 | 0.6662 | 0.3184 | 0.1050 | 0.0856 | | 1 | 5 | 4 | 3 | 2 |
| Max train Acc | 0.9888 | 0.7042 | 0.8808 | 0.9742 | 0.9938 | | 2 | 5 | 4 | 3 | 1 |
| Min train Loss | 0.0427 | 0.6654 | 0.3398 | 0.0943 | 0.0285 | | 2 | 5 | 4 | 3 | 1 |

Figure 5: Left Table: A quantitative comparison analysis was conducted on five optimizers. Values in each row are color-coded, where Dark Green represents the best value and Dark Red represents the worst value. The experiment consists of 480 images for training and 120 images for testing. The right table provides a ranking comparison of the five optimizers. Values in each row are ranked based on their performance and then color-coded, with Dark Green indicating the best value and Dark Red indicating the worst value.

0.0005, 0.001, 0.01, 0.1. The *Adam* optimizer, which has shown relatively good performance, is used for this experiment. The experiment follows the same setting as the previous experiment, employing cross-validation with five folds, and the results are averaged at each epoch over the five folds.

The following metrics are measured for each learning rate experiment:

1. Test Accuracy, (average and maximum).

2. Training Accuracy, (average and maximum).

3. Test Loss, (average and minimum).

4. Training Loss, (average and minimum).

5. $r$ (PCC), (average and maximum).

6. $p$_value, (average and minimum).

The objective of this experiment is to investigate whether changes in model performance (i.e., Accuracy and Loss) from one Learning Rate (LR) to another LR correspond to the same changes in r(PCC) and $p$-value.

Figure 7 (left Table) displays the numerical results for the 5 different Learning Rates, including accuracy (Training and Test), loss (Training and Test), r(PCC), and $p$-value. The data at each row is color-coded separately, with Dark Red indicating the worst performance and Dark Green indicating the best performance. Figure 7 (right Table) presents the ranked and color-coded values for better comparison.

Based on the results, the learning process with a Learning Rate of 0.1 exhibits the worst performance among all the experimented Learning Rates, which is also reflected in r(PCC) and $p$-value). In other words, the lowest Degree of Benford Law Existence (DBLE) is observed for the learning rate of 0.1. The second worst learning process is associated with a Learning Rate of 0.01, which is consistent with the DBLE (r(PCC) and $p$-value)).

The learning rates of 0.001 and 0.0005 demonstrate the best performance, and their DBLE aligns with these results. Finally, the learning rate of 0.0001 shows an average performance across all learning rates, and the DBLE also exhibits an average value. Therefore, based on this experiment with 5 learning rates, the model's performance is in accordance with the Degree of Benford Law Existence among the neurons' weights.

Figure 8 provide a detailed view of these 5 learning processes (5 learning rates), including Test and Train accuracy and Loss, as well as r(PCC) and $p$-value over 300 epochs. As the scale of the learning rates for 0.1 and 0.01 differs

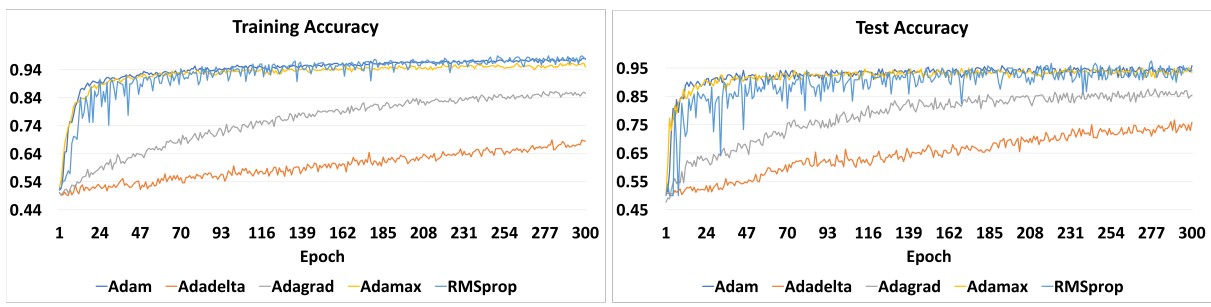

(a) The accuracy of Training dataset over 300 epochs for *PNEUMO-NIA* and *NORMAL* datasets. Note that all values are averaged over 5 folds.

(b) The accuracy of the Test dataset over 300 epochs for *PNEUMONIA* and *NORMAL* datasets. Note that all values are averaged over 5 folds.

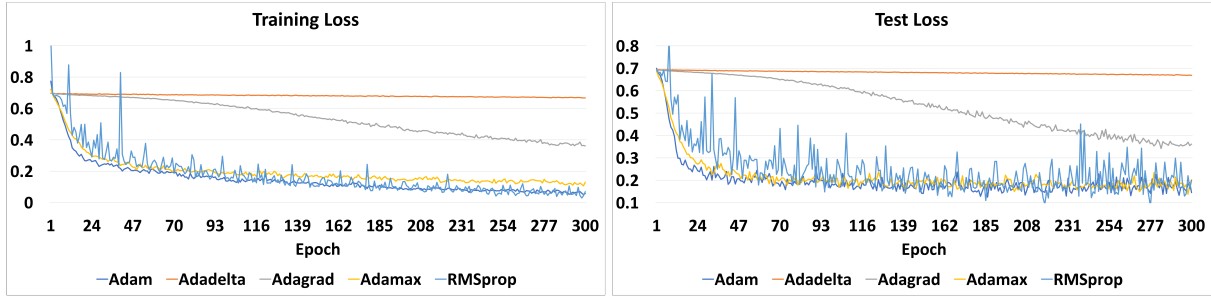

(c) The Loss of Training dataset over 300 epochs for *PNEUMONIA* and *NORMAL* datasets. Note that all values are averaged over 5 folds.

(d) The Loss of Test dataset over 300 epochs for *PNEUMONIA* and *NORMAL* datasets. Note that all values are averaged over 5 folds.

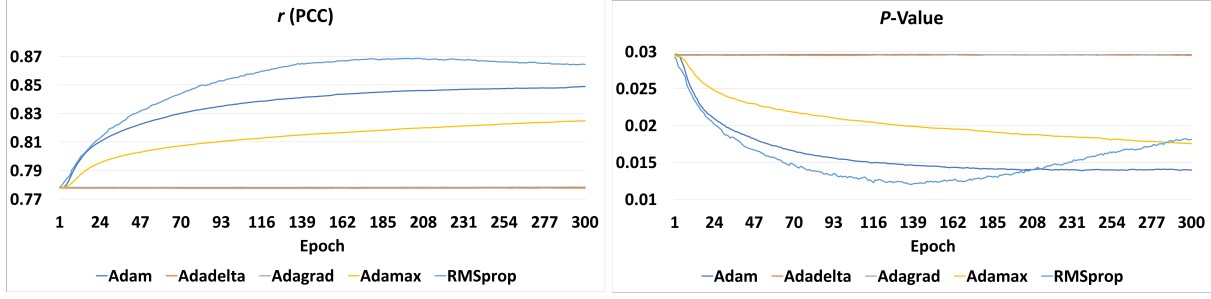

(e) The evolution of **r**-PCC on *BenList* and *BenIdeal* over 300 epochs for *PNEUMONIA* and *NORMAL* datasets. Note that all values are averaged over 5 folds.

(f) The evolution of *p*-Value on *BenList* and *BenIdeal* over 300 epochs for *PNEUMONIA* and *NORMAL* datasets. Note that all values are averaged over 5 folds.

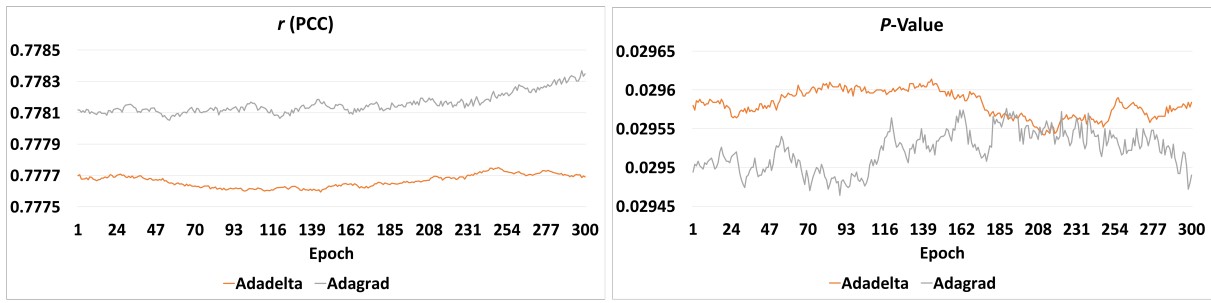

(g) Adagrad and Adadelta: The evolution of **r**-PCC on *BenList* and *BenIdeal* over 300 epochs for *PNEUMONIA* and *NORMAL* datasets. Note that all values are averaged over 5 folds.

(h) Adagrad and Adadelta: The evolution of *p*-Value on *BenList* and *BenIdeal* over 300 epochs for *PNEUMONIA* and *NORMAL* datasets. Note that all values are averaged over 5 folds.

Figure 6: The efficiency details of five optimizers for binary image classification on *PNEUMONIA* and *NORMAL* datasets with 480 images for training and 120 images for the test.

| Learnin Rate ==> | 0.0001 | 0.0005 | 0.001 | 0.01 | 0.1 | | 0.0001 | 0.0005 | 0.001 | 0.01 | 0.1 |
|---|---|---|---|---|---|---|---|---|---|---|---|
| Ave r (PCC) | 0.8197 | 0.8741 | 0.8546 | 0.6371 | 0.2575 | | 3 | 1 | 2 | 4 | 5 |
| Ave P-Value | 0.0203 | 0.0132 | 0.0191 | 0.1075 | 0.5450 | | 3 | 1 | 2 | 4 | 5 |
| Ave test Acc | 0.7998 | 0.8223 | 0.8149 | 0.5033 | 0.5001 | | 3 | 1 | 2 | 4 | 5 |
| Ave train Acc | 0.8413 | 0.9161 | 0.9131 | 0.4958 | 0.4918 | | 3 | 1 | 2 | 4 | 5 |
| Ave test Loss | 0.4427 | 0.6317 | 0.7669 | 0.6984 | 99.6198 | | 1 | 2 | 4 | 3 | 5 |
| Ave train Loss | 0.3582 | 0.1990 | 0.2083 | 1.2161 | 4126.3043 | | 3 | 1 | 2 | 4 | 5 |
| Max r (PCC) | 0.8527 | 0.8903 | 0.9021 | 0.6388 | 0.2638 | | 3 | 2 | 1 | 4 | 5 |
| Min P-Value | 0.0138 | 0.0090 | 0.0076 | 0.1064 | 0.5380 | | 3 | 2 | 1 | 4 | 5 |
| Max test Acc | 0.8800 | 0.8920 | 0.8980 | 0.6530 | 0.5170 | | 3 | 2 | 1 | 4 | 5 |
| Min test Loss | 0.3325 | 0.3706 | 0.3902 | 0.6467 | 0.6904 | | 1 | 2 | 3 | 4 | 5 |
| Max train Acc | 0.9425 | 0.9957 | 0.9967 | 0.6000 | 0.5400 | | 3 | 2 | 1 | 4 | 5 |
| Min train Loss | 0.1724 | 0.0200 | 0.0162 | 0.6633 | 0.6915 | | 3 | 2 | 1 | 4 | 5 |

Figure 7: Left Table: A quantitative comparison analysis was conducted on five Learning Rates. The values at each row are color-coded, with Dark Green representing the best value and Dark Red representing the worst value. The right table provides a quantitative comparison analysis of the five Learning Rates. The values at each row are ranked from the best one (1) to the worst one (5), and they are color-coded as well. Dark Green indicates the best value, while Dark Red indicates the worst value for each feature.

significantly from the scale of the learning rates for 0.001, 0.0005, and 0.0001, they are visualized in separate graphs to avoid visual overlapping.

## 4.3 Transfer Learning

One of the challenges in Machine Learning is the time required to train models, especially when dealing with large datasets. To mitigate this issue, emphTransfer Learning (TL) has been introduced (Torrey & Shavlik, 2010). TL involves using a pre-trained neural network model that has been trained on a large amount of data. When faced with a new training task, the model is not completely untrained, which can significantly reduce the learning time. Transfer learning is particularly useful in scenarios where the available data may not be sufficient to train a model effectively (Sheikh et al., 2020).

There are several well-known TL models available in different domains, such as Image Classification and Natural Language Translation. Some popular TL models in the Image Classification domain include *VGG16*, *Inception*, *Resnet*, *Xception*, *EfficientNet*, among others.

Given that TL models have already been trained on a large number of images, it is expected, based on the initial assumption of this work, that they exhibit a high Degree of Benford Law Existence (DBLE) among the weights of their neurons. To investigate this hypothesis, several TL models are loaded, and without any further training, the Benford List (BenList) is identified for each layer. The correlation coefficient (r(PCC)) and p-value between each BenList and the Benford Ideal (BenIdeal) are calculated, and the average values are reported. The details of the nine TL models can be found in Figure 11. It should be noted that each layer in the models may consist of multidimensional arrays, so the BenList is identified for all extracted 1-D arrays from each layer. Figures 9 illustrate the distribution of the weights according to Benford's Law on the accumulated pool of weights for each model. Figures 10 illustrates the curve created between BenList for each Transfer Learning model and BenIdeal.

As shown in Figure 11, all models exhibit high values for the correlation coefficient (r(PCC)) and a *P*-value less than 0.05, indicating that the null hypothesis (the BenList is unrelated to the Benford Ideal) can be rejected. It is worth noting that some studies (Abubakar et al., 2020) (Rahaman et al., 2020) have compared different Transfer Learning models; however, these comparisons are typically based on further training on new images. Therefore, it is challenging to directly compare these models against each other in the context of this experiment. Consequently, it is not possible to draw a definitive conclusion similar to the previous experiments. Nevertheless, the results presented in Table 11, Figure 9 and Figure 10 suggest that all the models exhibit a Degree of Benford Law Existence (DBLE).

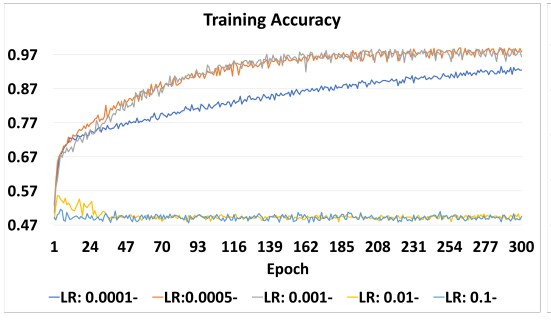
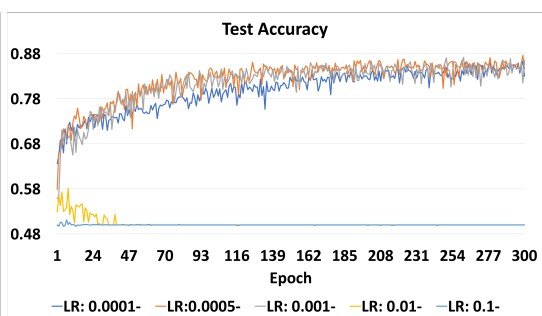

(a) The accuracy of Training dataset over 300 epochs for *moun-* *tain* and *sea* datasets applying Adam Optimizer. Note that all values are averaged over 5 folds.

(b) The accuracy of Test dataset over 300 epochs for *mountain* and *sea* datasets applying Adam Optimizer. Note that all values are averaged over 5 folds.

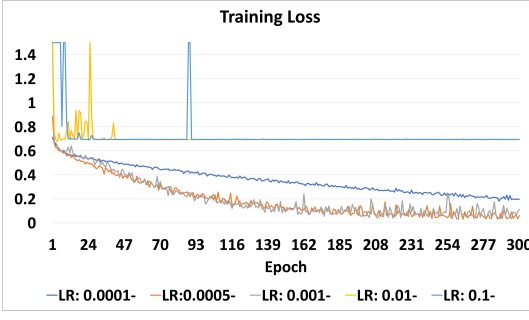
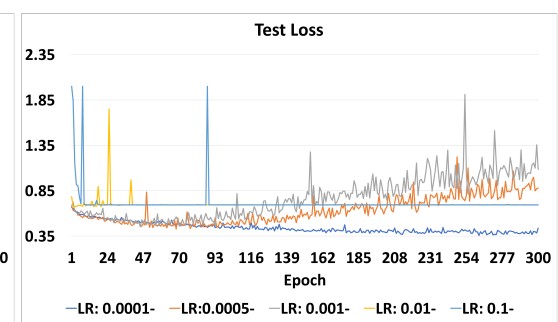

(c) The Loss of Training dataset over 300 epochs for *mountain* and *sea* datasets applying Adam Optimizer. Note that all values are averaged over 5 folds.

(d) The Loss of Test dataset over 300 epochs for *mountain* and *sea* datasets applying Adam Optimizer. Note that all values are averaged over 5 folds.

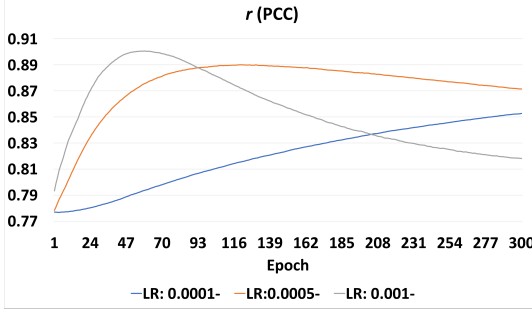
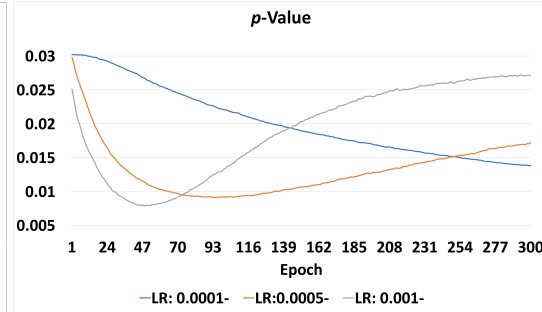

(e) The evolution of *r*-PCC on *BenList* and *BenIdeal* over 300 epochs for *mountain* and *sea* datasets applying Adam Optimizer. Note that all values are averaged over 5 folds.

(f) The evolution of *p*-Value on *BenList* and *BenIdeal* over 300 epochs for *mountain* and *sea* datasets applying Adam Optimizer. Note that all values are averaged over 5 folds.

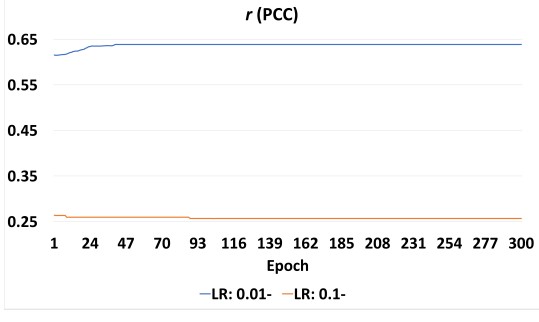
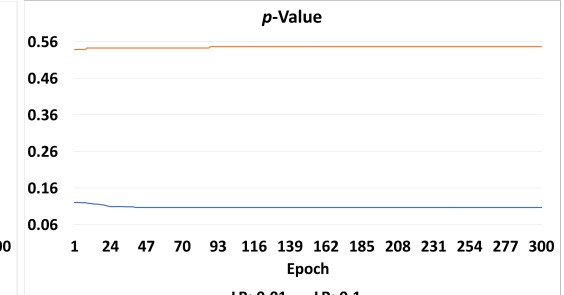

(g) LR: 0.1 and 0.01: The evolution of *r*-PCC on *BenList* and *BenIdeal* over 300 epochs for *mountain* and *sea* datasets applying Adam Optimizer. Note that all values are averaged over 5 folds.

(h) LR: 0.1 and 0.01: The evolution of *p*-Value on *BenList* and *BenIdeal* over 300 epochs for *mountain* and *sea* datasets applying Adam Optimizer. Note that all values are averaged over 5 folds.

Figure 8: The efficiency details of five Learning Rates for binary image classification on *mountain* and *sea* datasets.

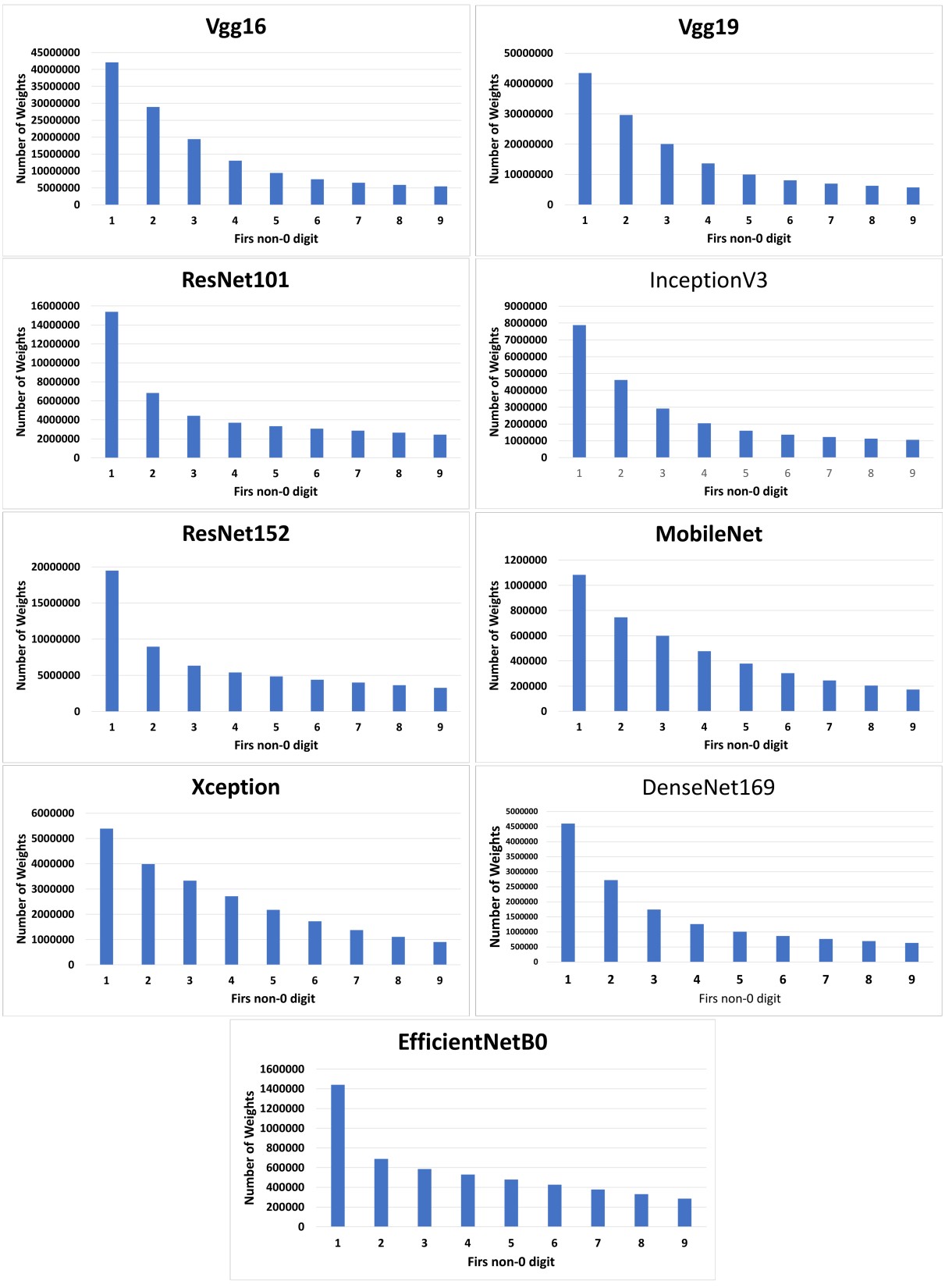

Figure 9: The details of 9 different Transfer Learning models from Benford-Law's point of view on the entire Neurons' weights.

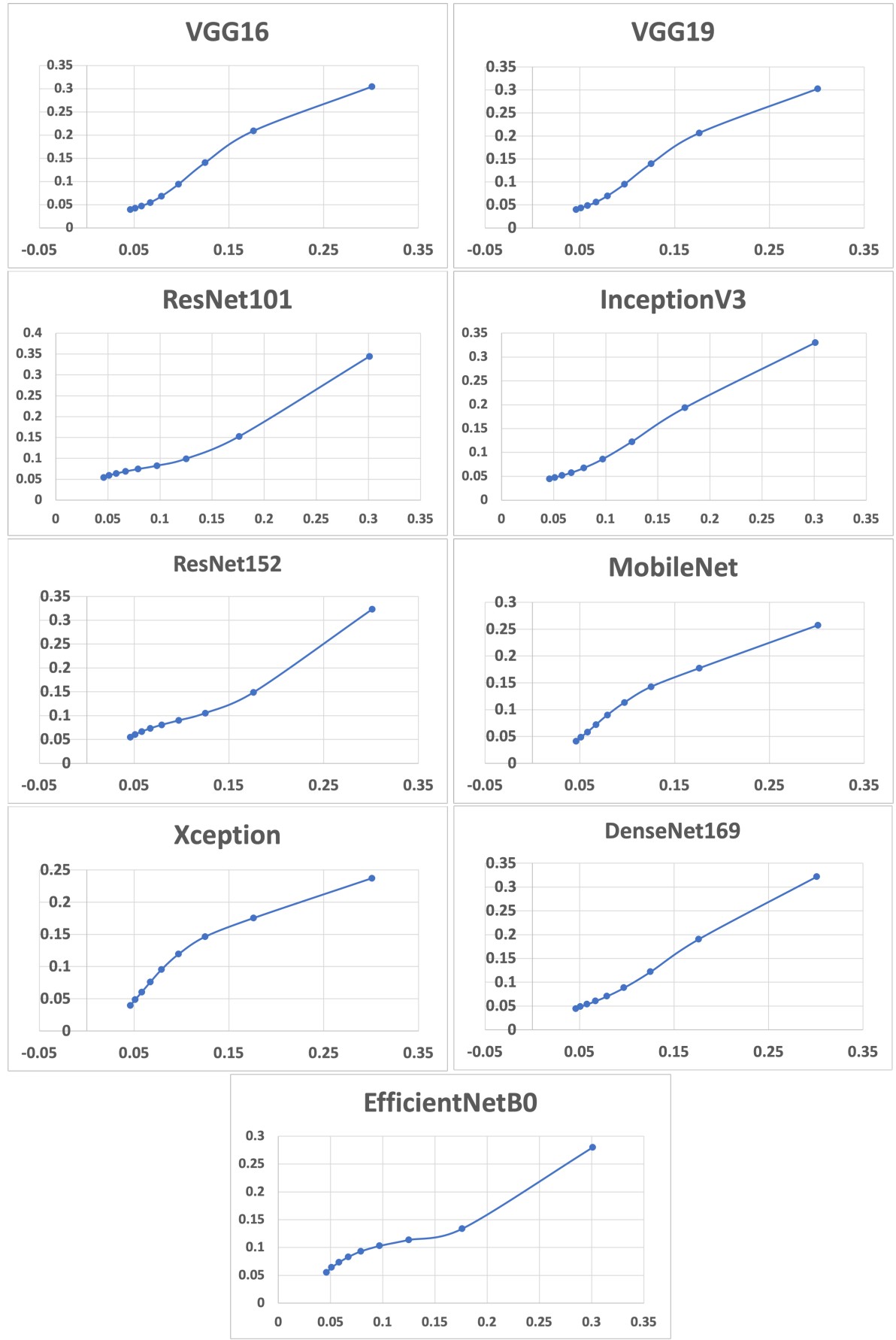

Figure 10: The Curve diagrams of 9 different Transfer Learning models from Benford-Law's point on BenIdea and BenList. The bar chart of the BenList for all these models are presented in Figure 9

| TL models | Ave r-(PCC) All 1-D Arrays* | Ave p-Value All 1-D Arrays* | Number of Total 1-D Arrays* |
|---|---|---|---|
| Vgg16 | 0.96025 | 0.00021 | 66731 |
| Vgg19 | 0.95884 | 0.00023 | 78254 |
| ResNet101 | 0.94873 | 0.00104 | 116276 |
| InceptionV3 | 0.94643 | 0.00199 | 96342 |
| ResNet152 | 0.94469 | 0.00115 | 170240 |
| MobileNet | 0.94356 | 0.00227 | 6120 |
| Xception | 0.93459 | 0.00143 | 26748 |
| DenseNet169 | 0.88034 | 0.01542 | 163928 |
| EfficientNetB0 | 0.83934 | 0.03771 | 21224 |

Figure 11: The details of 9 different Transfer Learning models are provided below. Please note that each model consists of multiple layers, and each layer may contain a multidimensional array. For each layer, all 1-D arrays are extracted, and separate calculations are performed for the correlation coefficient (r(PCC)) and p-Value. The final values are averaged. The models are listed in descending order based on their r(PCC) values, from best to worst.

## 5 Discussion

In this work, an experimental analysis is conducted to explore the connection between neural network neurons' weights and Benford's Law. To achieve this objective, three research questions are posed:

1. Do the weights of neurons in Convolutional Neural Networks conform to Benford's Law?

2. How do the weights of neurons change, from the perspective of Benford's Law, as the model undergoes training in a range of CNN models?

**Question1:** The weights in neural networks are typically generated using different methods. In Keras, the default weight generation approach is the uniform distribution using the *GlorotUniform* function. Uniform distribution alone often exhibits a degree of conformity to Benford's Law with an r(PCC) value around 70%. As discussed in sections 4.1 and 4.2, most of the learning processes demonstrate a high degree of conformity (DBLE) between the weights and Benford's Law, exceeding 70% in terms of r(PCC), with the exception of two learning rates: 0.1 and 0.01, where the weights fail to conform to Benford's Law. In the other experiment (see Section 4.3), nine transfer learning models are tested, and all of them exhibit a high degree of conformity between their neurons' weights and Benford's Law, as illustrated in Figures 9 and 11. Therefore, the answer to the first research question is **Yes**, with the exception of the two learning processes involving learning rates of 0.1 and 0.01, which will be further explored in the subsequent research questions.

**Question2:** It is evident that weights in neural networks change from one epoch to the next with the aim of improving model performance and reducing loss. This research question investigates how changes in weights are reflected in DBLE. As observed in the experiment conducted in Section 4, in most CNN models, the DBLE within the neuron weights tends to increase as the number of epochs grows. However, there are a few exceptions (i.e., the experiments with learning rates of 0.1 and 0.01) where the neuron weights do not conform to Benford's Law as the number of epochs increases. Considering that neurons are typically generated using a uniform distribution with an r(PCC) of around 70%, it can be concluded that the DBLE for all Transfer Learning models in Figure 11 has been increasing since the models started training. Therefore, the answer to the second research question is that *DBLE increases as epochs progress in most models*.

# 6 Conclusion

In this study, 15 models were designed and trained, and their details were recorded (refer to Figures 3, 5, and 7). In the first experiment, where different optimizers were tested with 1000 images, it can be observed that *Adadelta* performed the worst among the optimizers, and its DBLE exhibited the lowest values as well. *Adagrad* generally showed the second worst performance and had the second worst DBLE. The performance of *RMSprop* and *Adamax* were average among all optimizers, and the same applies to their DBLE (r(PCC) and *p*-value). Lastly, the *Adam* optimizer generally demonstrated the best performance, and the same applied to its DBLE (r(PCC)).

A similar pattern was observed in the other experiments that tested different optimizers with a lower number of images (600 images). In the experiment where different learning rates were tested with the *Adam* optimizer, learning rates of 0.001 and 0.0005 exhibited the best performance, which was also reflected in their DBLE (r(PCC) and *p*-value). The model with a learning rate of 0.0001 showed average performance among all learning rates, and its DBLE displayed average values as well. Finally, the learning rates of 0.1 and 0.01 performed the worst and second worst, respectively, and the same pattern was evident in their DBLE (r(PCC) and *p*-value). It is worth noting that changes in accuracy, albeit very small, indicate that the model is not halted or stuck in the middle of the learning process. Refer to Figure 8 for details.

Although this study was conducted using a limited number of models, the findings can serve as a guide for further analysis and future studies. This experiment represents the first attempt to analyze Neural Network weights from the perspective of Benford's Law. The experiments were conducted using one architecture with different settings and involved 9 transfer learning models. The initial results, serving as a proof of concept, have shown promising outcomes and confirm the existence of Benford's Law within Neural Network weights.

Based on the findings of this study, the initial hypothesis is shown to be valid, and it suggests that the distribution of neurons' weights tend to follow Benford's Law. This research can serve as a foundation for further investigations into the association between Benford's Law and Neural Network weights, opening up new avenues for research in this field.

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
