# OpenReview forum: "The Relationship Between the Distribution of Neural Network Weights and Model Accuracy Using Benford’s Law"
_TMLR — Rejected by TMLR_

### Review · Reviewer_H5Zo · 2023-10-29

**Summary Of Contributions:**

This paper presents an analysis of how Benford's Law relates to neural network weights. It aims to investigate the relationship between DBLE (Degree of Benford's Law Existence) and model weights in general, model weights over the course of training, and model performance. The paper proposes use of the Pearson Correlation Coefficient (PCC) to characterize the relationship between the actual distribution of first digits in weights (based on a transformation scheme from [-1, 1] to [1, \inf)) and the Benford's Law ideal.

The paper conducts a variety of experiments: binary image classification on 1,000 images with five different optimizers; binary image classification with 1,000 images, Adam optimizer, and five different learning rates; binary image classification on 600 different images with five different optimizers; and nine transfer models without extra training.

The results are shown in the form of heatmap tables over the PCC values and ranks for the dependent variables in each experiment, and line graphs showing evolution of PCC values and p-values over training time. For the transfer experiment, the transformed-weight-first-digit distribution is shown for all the trained models.

The paper finally claims that model weights do exhibit a level of DBLE that it considers high; that model weight DBLE increases over training; and that model accuracy is correlated with DBLE.

**Audience:**

Yes

**Claims And Evidence:**

Yes

**Requested Changes:**

- More thorough comparative analysis for the question of DBLE increasing over training time. Such analysis could also be applied to the question of DBLE being higher for more performant models, though those results are a bit more salient.
- Clearer motivation of this particular approach
- Discussion of the practical implications for machine learning practitioners

**Strengths And Weaknesses:**

#### Strengths
- Setup is clear and carefully done
- Paper is overall very well-written, and figures are very useful
- The results for the third question are convincing - for both the optimizer and learning rate experiments, it does seem that more performant models have higher DBLE.

#### Weaknesses
- This paper doesn't have clear motivation. At the end it concludes that the DBLE could be a good metric (proxy?) for the model's accuracy, but 1) this isn't spelled out clearly as the motivation at the beginning and 2) the related works and examples don't set up this motivation, as they are more about using Benford's Law alongside machine learning for separate purposes.
- The first and second question results aren't as convincing. The first question, whether model weights generally conform to Benford's Law, has a cutoff of 70% that isn't explicitly justified in the paper, even if it is a good result (I just don't know). For the second question, in many cases PCC does go up with training epoch, but in several cases it doesn't, particularly with learning rates (even performant learning rates). There isn't clear significance testing between these results and the actual margin of change in PCC is fairly small, so more thorough analysis would be helpful.
- If DBLE can be a good metric for accuracy, that is likely useful - it would be nice to have a way to assess a model's accuracy without having to run the test set (or at least in tandem as a descriptive statistic). However, the paper lacks clear quantitative frameworks for relating DBLE to accuracy - correlations are shown convincingly, but that doesn't entail a usable tool.

---

> ### Author Response · Authors · 2023-11-17
> **Comments and changes on the reviews**
>
> Thank you for the feedback.
>
> I've taken it into consideration for the revision. The paper didn’t aim to assert that DBLE would be a reliable metric for network quality, especially at this level of experimentation. As the text may create this assumption, I've revised it in the paper to eliminate this claim.
>
> Thank you for pointing out that the cutoff of 70% wasn't explicitly justified. Yes, that part was omitted, but it has now been included in the paper as a paragraph and a table, providing justification for the 70% cutoff in the methodology section. Regarding the second question, there are two learning rates that exhibit minimal change in PCC, indicating almost negligible improvements in training and test accuracy. To elaborate further on the second question, a brief paragraph has been added to the discussion section.
>
> The assertion about a stronger alignment between DBLE and model accuracy demands more extensive experiments. Hence, the third research question has been eliminated, framing this work as a proof of concept.
>
> The third research question is now omitted, while additional details regarding the first two questions have been included in the paper.
>
> A new paragraph has been incorporated into the methodology section to enhance the clarity of the motivation of the approach.
>
> The paper introduces a concept, Benford's Law in the context of Neural Networks, which hasn't been explored previously. Hence, the primary focus of this work has been on theorizing this concept or proof of concept rather than emphasizing its practical benefits. Nonetheless, as a proof of concept, this work lays the foundation for further analysis and exploration in this field.
>
> Thank you for the feedback once more. I hope the paper now presents itself in a more suitable form for potential publication.

---

### Review · Reviewer_sRxB · 2023-11-12

**Summary Of Contributions:**

This paper investigates whether Benford's Law exists in CNNs and whether it is related to the test error. The authors demonstrate the presence of Benford's Law for the weights of well trained neural networks, and show that this law gradually appears during the training.

**Audience:**

Yes

**Broader Impact Concerns:**

N.A.

**Claims And Evidence:**

No

**Requested Changes:**

Critical to acceptance: see Weakness (major) above;
Strengthen the work: see Weakness (minor) above.

**Strengths And Weaknesses:**

Strength:
1. The authors shows the presence of Benford's Law for well trained neural networks. This is an interesting result.
2. The authors shows that, in general, Benford's Law gradually appears throughout the training.


Weakness (major):
1. The provided evidences are not sufficient to support the generality of correlation between model performance and the DBLE. This is the strongest and the most useful result (if it holds) in this work. The authors should examine its validity more thoroughly to avoid providing misleading conclusion to other researchers. Performaning the following experiments could help. (a)  Systematically compare the DBLE of the state of the art pretrained models with the their test error to see if they are strongly correlated. I suspect not. From Figure 11, Vgg16 has higher DBLE to ResNet101. However, it is significantly less accurate than ResNet101. (b) Initialize the neural network from a distribution conforming with the benford law and trace the evolution of DBLE. (c) Using a sufficiently wide neural network and train it in a neural tangent kernel (NTK) regime. By the theory of NTK, the parameters stays at the vicinity of its initialization at the end of training, thus likely inherit the DBLE from their initial distribution.
2. No discussion about the mechanisms that could generate Benford's law, e.g., multiplicative fluctuation, and whether they may present during the training of CNN and could contribute to better generalization.

Weakness (minor):
1. Stochastic Gradient Descent is not considered in the experiments.
2. Other than the peason correlation coefficient, whether the slope of Benlist to BenIdeal cuve is close to 1 also indicates how good Benford's law holds. However, this slope is not provided throughout the paper.
3. No reference curve of BenIdeal to help the comparison in Figure 10.

---

> ### Author Response · Authors · 2023-11-17
> **Comments and changes on the reviews**
>
> Thank you for the comment.
>
> This paper didn't originally intend to claim that the network's performance could be solely evaluated by DBLE. To eliminate this assumption, adjustments have been made to the discussion section and methodology. As stated in the paper, the TL models were used without further training by us, making it challenging to determine their relative accuracy as it depends on various factors such as the dataset, among others. Furthermore, each non-TL model underwent training across 5 folds, which can somewhat mitigate biases. We agree that NTK would be a preferable choice for further analysis as this work is intended to serve as a proof of concept.
>
> While Benford’s law during training is not directly visualized, the r-PCC and P-Value between BenList and BenIdeal suggest its presence in CNN weights and during the training. The paper visualizes these aspects based on epochs, for instance, in Figure 5 e, f.
>
> There are certainly several other models that can be considered for this experiment, which will definitely be part of future studies.
>
> It's quite challenging and nearly impossible to present the curve from Benlist to BenIdeal as it's calculated for each individual epoch. Given that we have 300 epochs multiplied by 5 folds for each single experiment, it results in thousands of images to illustrate the curve from BenList to BenIdeal. Otherwise, it would have been an ideal metric as well. However, the appropriate reference curves have now been added to the paper for every single TL model. Thank you for pointing this out.

---

### Review · Reviewer_QoUm · 2023-11-13

**Summary Of Contributions:**

In this work the authors investigate whether Benford’s law applies to neural network weights, and how training and final network performance relate to changes in the degree of Benford’s law existence (DBLE). The authors show that ANN weights obey Benford’s law more than chance, and that optimizing network parameters tends to increase the DBLE for the network.

**Audience:**

Yes

**Claims And Evidence:**

No

**Requested Changes:**

Please see weaknesses.

**Strengths And Weaknesses:**

The paper is generally clear and well written and the topic is interesting. However, I think there are a number of weaknesses.

Weaknesses:

The most interesting and important claim of the paper, that DBLE can predict superior model performance, is not appropriately or critically assessed, and if anything, the experiment results indicate that this is not true (eg. compare Adadelta and Adagrad in Fig 4). In general I don’t think that the authors have applied appropriate levels of skepticism to this claim. For example final performance vs DBLE is not plotted or fit with a statistical measure. Furthermore, no theory or mechanism for why DBLE might be connected better performance is provided.

Related to the above, more generally as an empirical investigation, this manuscript would benefit from following the scientific method more closely. The 3 hypotheses are stated with no inductive reasoning from previous observations. What are the authors' intuitions or reasoning behind their hypotheses? This is important because such reasoning impacts experimental design and to build a more convincing empirical study. For example, if the mechanism by which DBLE increases during training is understood and empirically verified, it may be possible to link it to why DBLE might result in better performance. Then the experiments would be cohesive and build on top of each other.

In my opinion, the experiments with variants of sgd are not that useful for understanding the origin of changes in model DBLE w.r.t performance & training time, or at least are not analyzed. Why did the authors choose to explore Benford’s law and neural networks via the lens of different optimizers?

Minor weaknesses

- Benford’s law is introduced with a variety of examples, such as population statistics, finance, animal number etc. But there is no theoretical introduction or justification given which would help readers understand why Benford’s law should have any relation to ANNs or their training and performance.
- Figure 1 is repeated in figure 2, and figure text is too small.
- How is the corrcoef p value calculated (what are the samples here)?
- TL models aren’t assessed for DBLE vs final performance.

---

> ### Author Response · Authors · 2023-11-17
> **Comments on the reviews**
>
> Thank you for your comments.
>
> The paper did not aim to assert that DBLE can predict the superior model, especially in a study with a limited number of experiments. I have revised and rephrased the sections in the paper that might have conveyed such a claim. The third research question possibly led to that assumption, and it has been removed. Additionally, the discussion has been altered accordingly.
>
> I acknowledge that the third hypothesis may necessitate further investigations that are beyond the scope of this paper. Consequently, the third hypothesis has been rephrased and removed as a research question or hypothesis. This adjustment allows us to redirect the discussion towards better reasoning concerning the other two hypotheses.
>
> I comprehend the rationale behind the limited discussion of Benford’s Law from the optimizer's standpoint in the paper. Consequently, an appropriate exploration of its significance has been added to the Experiment section.
>
> The main motivation arose from the comparison of neural network (NN) populations to data sets like countries’ populations and financial figures. Despite conducting an extensive literature review on the applications of Benford’s Law, we found no prior experiments involving the application of Benford’s Law to neural network weights. Hence, this study aims to serve as a proof of concept, paving the way for further analysis and research in the future.
>
> The redundant images have been removed, and the text has been enlarged.
>
> The formula for calculating Corrcoef has been included in the paper. Additionally, a reference to the Python implementation for computing r and p-Values has been provided.
>
> Although there is no visualization directly comparing DBLE versus Final Performance, tables are available for such comparisons, such as Figure 8, Figure 6, and Figure 4. These tables detail the p-value and r-PCC concerning Max training and Test accuracy and loss.
>
> I hope the revision will enhance the paper's quality for potential publication.

---

### Decision · Action_Editor_kfuK · 2023-12-20

**Recommendation:** Reject

**Comment:**

The reviewers felt that the strongest claim of the paper was regarding the correlation between DBLE and model performance, but all three noted issues with the empirical evidence for this claim.  The authors responded by toning this down, but then the reviewers were left feeling that there was no conclusion or implication derived from the observation in the paper.

In particular, reviewer QoUm noted that "it is still unclear to me what to take from the paper. I do not think that, given the results presented, that DBLE can be used to predict superior model performance (and the authors do not intend to make this claim). Also, for generalizability, I think it is very important to present a theoretical understanding of why and when networks exhibit benford's law or not, or to empirically attempt to understand this question (e.g. what happens when the data does not follow benford's law?). Overall, I think this is an interesting start, but further work is needed to strengthen the submission."

Further, reviewer H5Z0 states that "other reviewers seemed to agree with me that the paper suggested that DBLE could be a good proxy for/measure of model accuracy but didn't prove this - in fact, the other reviewers are even less convinced than me that there is any meaningful relationship, let alone one that makes it a good metric. The responses/revision mainly say that that wasn't a goal of the paper, and it was just initial exploration. I felt that of all the three research questions, that was the only one the paper made any progress on; beyond that, it's unclear what the key findings are."

In the end, two of the three reviewers felt that the paper is not ready for publication at TMLR.

**Audience:**

I am not convinced that the audience would find this interesting in its current state.  The issue here is that showing a correlation between the weights of a CNN and Benford's Law does not have a clear implication that is articulated by the authors.  More relevant would be either to show this in a more general setting, or to analyze why this is happening, or to discuss why this is an important implication.  Does this have implications to training models?  Constructing architectures? Why does the correlation increase with training?  There are potentially interesting questions here to pursue but right now it feels a little half baked.

**Claims And Evidence:**

Initially, one of the paper's claims that all three reviewers took issue with was that DBLE can predict superior model performance.  This was not supported well in the paper, as noted by all three reviewers.  Ultimately, the authors toned down this claim from the paper.

The main claim now is that Benford's Law does show up during training CNNs, and further that the DBLE increases during training in many cases.  This is supported by evidence.

However, I do note that, if one is going to write a paper on the correlation between the weights of a CNN and Benford's Law, one needs to delve a little deeper on why it is happening, or what it means.  In terms of missing evidence, it is unclear how general this phenomenon is, and whether or not there is an explanation for it, mathematical or otherwise.

**Resubmission Of Major Revision:**

The authors may consider submitting a major revision at a later time.